# Implementing Pedagogical Approaches for ESD in Initial Teacher Training at Spanish Universities

**Sílvia Albareda-Tiana** [1],*, **Esther García-González** [2], **Rocío Jiménez-Fontana** [2] **and Carmen Solís-Espallargas** [3]

1   Faculty of Education, Universitat Internacional de Catalunya, 08017 Barcelona, Spain
2   Faculty of Education, Universidad de Cádiz, 11519 Cádiz, Spain
3   Facultad de Educación, Universidad de Sevilla, 41013 Sevilla, Spain
*   Correspondence: salbareda@uic.es; Tel.: +34-932541800

**Abstract:** Within the framework of the United Nations 2030 Agenda for Sustainable Development, four case studies of the degree in Primary Education at three Spanish universities are analyzed. The aim is to study the suitability of three different active teaching-learning strategies: problem-based learning (PBL), project-oriented learning (POL), and a cross-disciplinary workshop. Another goal is to promote the integration of education for sustainable development (ESD) and measure the level of acquisition of several competencies of sustainability and the change in consumption habits of future teachers after implementing those pedagogical approaches. Initial and final ecological footprint (EF) as well as a rubric to measure the level of acquisition of competencies of sustainability were used as data collection instruments. The conclusions related to the research objectives show that when sustainability is implemented in the curriculum through active teaching-learning strategies, future teachers acquire competencies of sustainability. They also reveal that said strategies contribute to a change in consumption habits as a reduction in the EF is observed. There exists a relation between EF reduction and high levels of acquisition of competency in sustainability.

**Keywords:** initial teacher training; sustainable consumption patterns; education for sustainable development; individual ecological footprint; transformative learning; sustainable competencies

## 1. Introduction

Fridays for Future, the recent youth movement inspired by the actions initiated by Greta Thunberg to act against climate change, has led to weekly concentrations of young people in over 100 countries. This movement is awakening awareness among citizens and shows how young people are able to commit themselves and lead sustainability projects.

Problems such as climate change and biodiversity loss cannot be prevented if there is no change in consumption patterns. This requires a change in the way we think and act [1] with regard to consumer habits. Training university students to adopt sustainable behavior is crucial. University education is therefore not only necessary to transmit knowledge, but also to provide future graduates with competency-based training [2,3].

Intense debate exists regarding the terminology surrounding sustainable development, sustainability, environmental education (EE), education for sustainability, and education for sustainable development (ESD). According to Calvo and Gutiérrez [4], ESD has become the most widely used in the context of environmental education as it defends a complex and integrating approach in the curricula in all areas and at all levels of education. This integrated vision of ESD is in line with the global vision of sustainable development proposed by the 2030 Agenda [5] and the reports published by UNESCO for its educational implementation in education [2,6].

Competency-based teaching and learning was introduced in universities when the European Higher Education Area was implemented. In accordance with the Bologna Declaration, this study considers the concept of competency as the capacity to integrate knowledge, skills, and attitudes in order to perform tasks and solve problems in a given context [7].

In less than two decades, numerous initiatives have been taken to promote competencies of sustainability in higher education (HE) [8–12]. Teaching based on competencies of sustainability therefore presents a new challenge for university teachers and, more particularly, for those who teach future teachers [13–18].

It is not enough for teachers to know about the problems related to sustainability, they need to have acquired the necessary competencies of sustainability and teach the way they live [14]. The UNESCO reports for ESD consider addressing sustainability in an interdisciplinary manner as a challenge in teacher training for ESD and recommend integrating sustainability into the curriculum, thus encouraging future teachers to acquire competencies of sustainability [19,20]. Over the past decade, numerous initiatives in this field have been undertaken. The UNESCO publication Issues and Trends in ESD includes different case studies in teacher training on ESD (see [19] pp. 96–105). However, "despite international recognition of the significance of teacher education as a means to advance the status of ESD worldwide, there is still a need to mainstream learning for sustainability into pre-service teacher education in a consistent manner" [19] p. 147.

This paper includes case studies carried out by four teachers in the subjects of Didactics of Science (DofS) and Environmental Education (EE) in the primary education degrees of three Spanish universities. The emerging frameworks for teaching ESD recommended in the latest UNESCO reports [19] (pp. 148–149) and UNESCO's Global Action Programme on ESD [21] for teachers were addressed in those subjects. They can be summarized as follows: (a) *Learning content*: integrating critical issues, such us sustainable consumption, into the curriculum; (b) *Teaching and learning in an interactive learner-centred way*, using PBL and POL strategies, (c) *Promote competencies on ESD* and (d) *Societal transformation:* empowering learners (future teachers in this case) to be transformed and to transform the society they live in [21].

With regard to the learning content in these case studies, it seemed appropriate to include contents related to several sustainable development goals (SDGs) [6] as they are comprehensive enough to be addressed from a holistic viewpoint using active didactic strategies.

The SDGs include a complex range of social, ecological, and economic challenges such as reducing poverty, mitigating climate change, and promoting sustainable consumption, which must be addressed comprehensively. Universities must play a leading role in order to bring about change towards fairer and more sustainable societies. However, the question is how to integrate the SDGs into the university curriculum for education to be transformative rather than merely informative [22,23].

The paradigm shift that achieving the SDGs implies can only occur in our societies through education and learning [6]. It is not only about knowing the problems the planet and its inhabitants are experiencing, but about being able to collaborate in finding solutions for the SDGs and this can be an opportunity to mobilize youth. Sustainability needs to be addressed in a global manner, not only from an ecological viewpoint. The case studies here described deal with problems or projects related to SDGs 3, 6, 12, and 15 but in all four case studies, the students' ecological footprint, which is directly related to SDG 12: sustainable consumption, was measured.

In the literature, several research studies on education for sustainable consumption as an important part of ESD can be found. Resource consumption is directly related to environmental deterioration. However, human behavior has hardly changed in recent years. It is therefore necessary to insist on sustainable consumption at all levels through education [24]. Education for sustainable consumption is poorly developed in higher education. It should be included in the curriculum as universities are responsible for training citizens and need to promote sustainable consumption patterns [25]. In this study, the students' competencies in sustainable consumption were not measured specifically [26–28]. What was examined was whether there was a change in the students' consumption habits after

implementing the pedagogical approaches. Education is key to encourage reflection and promote consumption habits that involve significant changes in the way resources are used. ESD in initial teacher training is therefore essential, as those future teachers are a reference to society. If they have acquired sustainable patterns, they can encourage sustainable behavior changes in future generations.

Given the prevailing need of transforming teacher education to promote sustainable futures [19] (p. 138), in initial teacher training, the teachers of the abovementioned subjects analyzed the suitability of the different learning-teaching strategies promoting the integration of ESD. It was also an opportunity to contribute to empowering young people within the framework of the SDGs [6].

The purpose of this paper is to measure the level of acquisition of certain competencies of sustainability and the change in consumption habits of future teachers after the teaching-learning strategies were implemented.

Future teachers need to be empowered to be global citizens who engage and assume active roles to solve sustainable challenges and become proactive contributors to creating a more sustainable world [19].

To this end, the following research objectives are proposed:

1.  Analyze, through comparison, whether the active teaching-learning strategies contribute to a change in the consumption habits of future teachers and measure it through the individual EF of the students.
2.  Analyze whether there is a relation between the students' reduction in some of the fractions of the EF and the acquisition of competencies of sustainability after implementing the active teaching-learning strategies.

## 2. Materials and Methods

According to Rieckmann, 'transformative learning can be defined primarily by its aims and principles, not by a concrete teaching or learning strategy' [29] (p. 49). In this study, following the objectives, teaching-learning strategies have been implemented including characteristics common to pedagogical approaches for ESD that allow moving from knowledge to action [21]. The same rubric for evaluating competencies of sustainability was used in all the case studies performed in initial teacher training [30].

### 2.1. Participants

The case studies were developed during the first four months of the 2018–2019 academic year. A total of 93 students of the Degree in Primary Education at the following universities: Universitat Internacional de Catalunya (UIC), Universidad de Sevilla (US) and Universidad de Cádiz (UCA) participated. The participants were duly informed and gave their consent regarding collecting their data. The subjects involved in the project were: Didactics of Science (DofS) (three groups) and Environmental Education (EE) (one group). Sixty-nine students (74.2%) were girls and 24 (25.8%) were boys (Table 1).

**Table 1.** Organizational context and structure of the four case studies in Spanish universities.

| University | US | UCA | UCA | UIC |
|---|---|---|---|---|
| **Degree** | **PE** | **PE** | **PE** | **PE** |
| Level | 2nd year undergraduate students | 3rd year undergraduate students | 4th year undergraduate students | 3rd year undergraduate students |
| Credits | 6 credits | 2 credits | 6 credits | 6 credits |
| Subject | DofS | DofS | EE | DofS |

**Table 1.** *Cont.*

| University | US | UCA | UCA | UIC |
|---|---|---|---|---|
| **Degree** | **PE** | **PE** | **PE** | **PE** |
| Academic course | | 18–19 | | |
| Teaching-learning strategies | Problem-Based Learning (PBL) and Cross-Disciplinary Workshop (C-DW) | Problem-Based Learning (PBL) and Cross-Disciplinary Workshop (C-DW) | Project-Oriented Learning (POL) | Project-Oriented Learning (POL) Cross-Disciplinary Workshop (C-DW) |
| No. students | 40 | 13 | 23 | 17 |

*2.2. Data Collection Instruments*

In this research, two data collection instruments were used: (a) the students' EF and (b) an instrument to measure the students' level of acquisition of competencies of sustainability through a specific rubric for initial teaching training [30].

2.2.1. Description of the On-Line Calculator Selected and Data Analysis

The online tool myfootprint.org was used as an instrument for calculating individual EF.

Consumption measurement tools known as "footprint indicators" are used to examine the human demand for renewable resources and ecological services. Through micro and macroeconomic systems, they help to illustrate the relationships between human beings and the environment. They are expressed in global hectares of required productivity. The understanding of social and economic factors and their environmental impacts can guide decision-making aimed at strengthening sustainability [31].

The EF is an index on how the lifestyles of individuals or societies affect the environment, expressed as the area of productive land required to satisfy the consumption of natural resources and to assimilate the waste generated during a year. It is expressed in global hectares (gha) per person per year [32].

Since the EF was designed as an indicator that enables measuring the level of consumption of individuals and countries in terms of physical space [32], it has been used as a pedagogical tool in different university studies [1,14,33].

It has advantages over other calculators because it can be adapted to the country in which individuals reside, providing an average figure per inhabitant in hectares (ha), as well as a global calculation per planet, which refers to the number of planets that would be necessary to support all the people that live on the planet in a given year. The tool also offers an EF calculation for each of the following indicators or fractions: carbon footprint (CF), food footprint (FF), goods and services footprint (GSF), and housing footprint (HF). In addition, it provides information and recommendations on the most sustainable alternatives of each one. Although it has limitations like other online tools since its calculation is quick and provides insufficient information about its methods [33], in educational research [34] it is useful because it enables transforming consumption habits of students, from qualitative to quantitative data and making comparisons about changes and how actions directly affect the environment [35]. The EF is the most commonly used instrument to calculate consumption [33,34,36]. Although it focuses on ecological impacts, it is derived from socio-economic ones since socio-environmental problems cannot be broken down; they are systemic and global.

Even though there have been changes in behavior towards more sustainable lifestyles in recent decades, their scope and impact are not easily identifiable as the changes required to make progress with regard to sustainability remain complex and extensive. Until now, most of the measurements have been related to household $CO_2$ emission levels, without considering other changes related to human development, health, governance, collaboration, etc. [37]. Currently, there are no tools that measure these more sociable characteristics of sustainability.

In this study, the EF of each student was measured before and after implementing the pedagogical approaches, which had an average duration of five months.

2.2.2. Description of the Rubric for Assessment of Competencies of Sustainability

This rubric was developed in the EDINSOST research project [30], in which a map of sustainability competencies was designed for education degrees and postgraduate studies. Researchers from ten Spanish universities took part in the project [30].

The general goal of the EDINSOST project is to advance in education innovation for sustainable development in universities, to provide future graduates with the necessary competencies to promote change towards a more sustainable society.

The rubric includes four generic competencies of sustainability, previously approved by the Conference of Rectors of Spanish Universities (CRUE) [11] to be implemented in a cross-curricular manner in all the degrees of the Spanish University System (SUE).

These competencies are organized in levels of acquisition following Miller [38], who established a hierarchy of competencies in the medical profession (that can also be applied to other professions) which are depicted in Figure 1.

Miller established four skill acquisition levels defined by learning outcomes (indicators) based on the standards established for the National Centre for Education Statistics in the United States [39]. In this research, the version of the EDINSOST project in which the two higher levels of competencies are put together was used. The first level (KNOWS) at the base of the pyramid corresponds to knowledge and refers to learning, the second level (KNOWS HOW) corresponds to integration and development in a situation, and the third level (SHOWS HOW & DOES) is related to showing competency during action and the possibility of transferring that action.

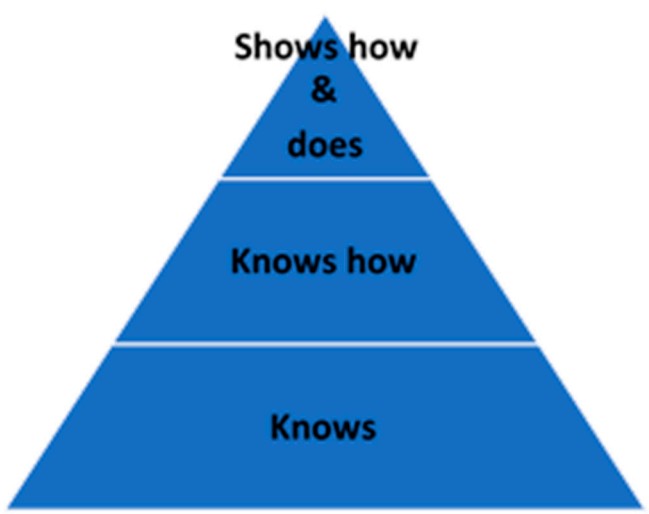

**Figure 1.** Miller's Pyramid (1990).

The competency map enables evaluating the results of learning acquired by students in those three levels. Competencies 1 and 3 were chosen as competencies common to all the case studies in this research (Table 2).

**Table 2.** Competencies 1 and 3 of the competency map on sustainability in education at the university level (Albareda et al., 2018) within the framework of the EDINSOST project.

| | SUST Competency Map of All the Degrees in Education | | |
|---|---|---|---|
| **Related Competencies** | **Acquisition Levels (According to the Simplified Miller Pyramid)** | | |
| | **Level 1: Knows** | **Level 2: Knows How** | **Level 3: Shows How & Does** |
| SUST 1—Critical contextualization of knowledge establishing interrelationships between social, economic, environmental, local, and/or global problems | Knows the functioning of natural, social, and economic systems and the mutual relations between them | Analyzes and understands the relationship between natural systems and social and economic systems | Is able to imagine and predict the impacts the changes produced in natural systems may cause in social and economic systems and among each other |
| SUST 3—Participation in community processes that promote sustainability | Recognizes himself/herself as an integral part of his/her surroundings and knows the community education programmes that encourage participation and commitment to socio-environmental improvement | Is able to interact satisfactorily in educational community projects, encouraging participation | Designs and carries out socio-educational activities in participatory community processes that promote sustainability |

### 2.2.3. Pedagogical Approaches for ESD

The common characteristics of all the pedagogical approaches for ESD in initial teacher training are the UNESCO Roadmap recommendations for implementing the Global Action Programme (GAP) on ESD [21]. Table 3 shows the common characteristics of the different pedagogical approaches for ESD.

**Table 3.** Common characteristics of pedagogical approaches for education for sustainable development (ESD).

| University | Learning Content in the Curriculum (Related to SDGs) | Teaching and Learning in an Interactive Learner-Centred Way | Promote Competencies of ESD | Empowering Learners to Transform Themselves and the Society They Live in |
|---|---|---|---|---|
| UIC | SDG 12 | POL | SUST 1 and 2 | Cross-Disciplinary Workshop |
| US | SDG 12 | PBL | SUST 1 and 2 | Cross-Disciplinary Workshop |
| UCA_EofS | SDG 3, 6, 12 | PBL | SUST 1 and 2 | Cross-Disciplinary Workshop |
| UCA_EE | SDG 15 | POL | SUST 1 and 2 | Didactic materials for the local environment |

The instrument presented in Table 4 was used to assess those competencies.

**Table 4.** Instrument for assessment of two competencies of sustainability approved by the Conference of Rectors of Spanish Universities (CRUE) using the levels of competency of the simplified Miller pyramid.

| Sustainability Competencies Approved by the CRUE 2012 and Adapted for Education by EDINSOST 2018 | Lacks Command of the Competency | Level of Acquisition of the Competency According to Miller | | |
|---|---|---|---|---|
| | | **Knows** | **Knows how** | **Shows how and Does** |
| **Level** | **0–0.5** | **0.5–1** | **1–1.5** | **1.5–2** |
| 1. Understands the functioning of natural, social, and economic systems, as well as their interrelations and problems, both at a local and global level | | | | |
| 3. Promotes and participates in community activities that encourage sustainability | | | | |

## 3. Case Studies: Transformation in Learning and ESD (Teaching-Learning Strategies Used)

*3.1. POL and a Cross-Disciplinary Workshop on Sustainable Consumption at the Universitat Internacional de Catalunya (UIC)*

### 3.1.1. Initial Phase

In small groups, the students had to develop a research project on a real problem related to the consumption aspect of SDG 12 as a method to develop competencies of sustainability. They worked

on the projects in the subject of Didactics of Science. Through POL, a shared cocreation process was performed in a learning environment that fostered research.

### 3.1.2. Pedagogical Approach Implementation

Using the POL strategy, several groups of four students engaged in a research project related to sustainable consumption for a period of two and half months. The topics of the research project were the following: Do you know what polyethylene is and what its impact is if it is not recycled? Are we aware of the use of aluminium? Water bottle caps, a new life for solidary plastic caps; Responsible production, consumption, and use of drink cartons; recycling of canned food; Is the production behind a large brand such as NIKE sustainable?

During this period of research and interactive reflection, POL tutorials were offered to the students and they contacted stakeholders outside the university such as public administration and experts on sustainable consumption in different sectors and schools. In this phase, the groups of students learned in a formative and participatory process.

### 3.1.3. Final Phase: Cross-Disciplinary Workshop on Sustainable Consumption

The students presented their research findings using a scientific poster or an interactive workshop to the rest of the university community. It encouraged addressing sustainable consumption in a cross-disciplinary manner. After presenting their projects, they were evaluated by a group of experts. Twenty experts in the subject, including lecturers from the university itself, lecturers from four other universities, and experts from the Sustainability Department of Barcelona's City Council participated in the evaluation of the projects. All the evaluators had the same data-collecting instrument for the assessment of the students and they assessed the following competencies: (1) Understands the functioning of natural, social, and economic systems, as well as their interrelations and problems, both at a local and global level and (3) Promotes and participates in community activities that encourage sustainability.

### *3.2. PBL at the Universidad de Sevilla (US)*

This strategy was implemented in the subject of Didactics of Science. One of the objectives of this subject focuses on the recognition of the mutual influence between science, technology, society, and the environment, as well as appropriate citizen behavior in order to achieve a sustainable future, which is related to the consumption aspect of SDG 12 (Table 3).

### 3.2.1. Initial Phase

The first phase revolved around the awareness of the socio-environmental issues that most concern students and their possible causes. After a group debate on the ideas students had about the possible consequences of environmental problems, we focused on consumption. For that purpose, the activities were aimed at answering two work questions: (1) where do the objects we buy come from and where do they go? and (2) Do we need everything we buy?

### 3.2.2. Pedagogical Approach Implementation

This phase focuses on examining the main strategies that encourage conscious and unconscious consumption. To that end, the students were asked to carry out a research project based on the question: Why do we buy more than we need? On one hand, students performed a field study in which they collected on-site data on the strategies used by sales areas (supermarkets, clothing or shoe shops, large hotel chains, etc.) to promote consumption. On the other hand, they analyzed what strategies advertising uses to encourage consumption and how they could be used for counter-consumption. The students gave a presentation of the results of their research. After sharing their conclusions,

a debate was generated in which the students agreed on the main strategies used by advertising to promote consumption.

### 3.2.3. Final Phase

The purpose of this phase was to establish an action plan to raise awareness of the socio-environmental problems generated by excessive consumption. The students were asked to design a project aimed at young people towards more conscious and sustainable consumption. The students proposed an awareness campaign for young people between 18 and 30 on Instagram. To do this, several work groups designed the most appropriate strategies for the campaign to be successful. As a result, 13 projects were developed: 10 videos, one poster, and one survey whose aim was to raise awareness about more responsible and sustainable consumption. An Instagram account was opened: @alconsumonomesumo_ and the campaign started a few days before Christmas. Including social networks has several advantages: greater and better access to a young audience, survival of the project over time (as long as feedback is maintained), opinions can be collected through the comments of the recipients (which allows collecting feedback from the projects), speed and immediacy of access, unlimited reproduction, and a domino effect through inviting followers and forwarding. Action goes beyond the university classroom.

For the assessment of the level of competencies acquired in sustainability, different instruments for the assessment of two competencies of sustainability were used (Table 3). To assess the change in habits, a pre- and post-test questionnaire on the EF was carried out.

### 3.3. PBL and Cross-Disciplinary Workshop (Congress) at the Universidad de Cádiz (UCA)

### 3.3.1. Initial Phase

The students were asked to plan and design a research process with the aim of increasing their knowledge. For that purpose, the teacher offered several topics of natural sciences framed in the primary curriculum and each team (4–5 students) chose one. Some guidelines were provided: TASK 1: Analyze the problem scenario. Verify the understanding of the scenario through discussion with the team. TASK 2: What do we know about the topic? Brainstorm. Make a list of hypotheses and possible theories on the topic in question. TASK 3: What do we want to know? Make a list including everything that is unknown and everything we think we should know to solve the research around the topic. TASK 4: How do we specify the research problem? Explain in a couple of sentences what the team wants to solve, produce, answer, test, demonstrate, etc. TASK 5: How do we plan the research? Define activities, roles, and function of each one, how the information will be obtained, from what sources, using what instruments, sequencing, timing, etc. TASK 6: Presentation of the provisional research scheme.

### 3.3.2. Pedagogical Approach Implementation

During this phase, the students had to carry out their research. They worked independently and had to develop the theoretical framework sustaining the research, design the data collection instruments, find the empirical sources, analyze the data obtained, and solve the research problem initially proposed. The topics selected by the students were "*We all want coverage*", "*Does water have an owner?*" and "*Plasticized*", related to SDG 3 (*Good health and well-being*), SGD 6 (*Clean water and sanitation*), and SGD 12 (*Sustainable consumption*) respectively (Table 3). Once the research was conducted, they had to design a scientific poster to be able to participate in the second Congress of Primary Education Teacher Training in Didactics of Science (MAFEPRID-CN 2019).

### 3.3.3. Final Phase

This phase corresponded to the congress, in which a total of 180 third-year students of the degree in Primary Education and a scientific committee formed by teachers from the Didactics of Science department participated for a week.

To assess the congress, different data collection and analysis instruments were used: group portfolios, oral presentations, self-evaluation and coevaluation sheets, final product, teachers' diaries, a rubric for the evaluation of the scientific posters, final rubric, and sustainability competencies map. The teachers of the subject, the scientific committee, and the students themselves took part in this evaluation.

### 3.4. Participative ESD Project with the Local Community in a Nature Park, Universidad de Cádiz (UCA)

#### 3.4.1. Initial Phase

Different activities were developed, such as individual and group questionnaires, outings to places in the local environment, debates, reading sessions, presentations, and a group discussion of the global environmental crisis. The aim was for the students to acquire a complex vision of sustainability and to familiarize them with the main environmental issues. This phase prepared the students for the next phase and provided them with greater knowledge about the topics (socio-environmental problems and possible solutions, potential of ESD in primary education) to be worked on in the Environmental Education subject.

#### 3.4.2. Pedagogical Approach Implementation

The main objective of this phase was to provide students with an experience based on project work. The theme chosen was *El Parque de los Toruños*, a natural space where the Faculty of Educational Sciences is located. A common project was designed in the class group including six different research lines. The students were organized into groups of 5–6 members. In this phase, the students worked autonomously, following a series of guidelines proposed by the teacher: analysis of knowledge about the chosen line, necessary training needs, search for information, and designing and developing a final product. The following were included in this phase: field trips, search for information (bibliography, webography, interviews with experts), structuring and synthesizing information, oral presentations, coevaluation, group evaluation, etc.

#### 3.4.3. Final Phase

The work developed by the different groups was presented. The idea was to share the knowledge worked on during the course both with the university community and with local citizens.

The information the students handled was materialized in different products, starting from the premise that the project had to have a specific impact on the local surroundings. The following were used: a scientific poster presented in the faculty about the developments in the natural park; an open blog about the park and its people; a lapbook on environmental issues presented at a primary school; a model on the type of soil, birds, and plants of the park also presented at a primary school; a documentary about the civilizations that have inhabited the park; and a dossier with possible activities to develop in the park related to life on land (SDG 15) (Table 3) that was sent to the managing bodies of the park. In short, they carried out an in-depth analysis of the connections between the different socio-environmental aspects of sustainability and the impact of consumption on the environment.

To assess the subject, different data collection and analysis tools were used: journals for the group work follow-up, oral presentations, coevaluation sheets for critical analysis of the knowledge shared in the presentations, a final product that makes an impact on the local environment, an individual report on the learning acquired, and the teacher's diary in which the items related to the acquired competencies of sustainability and the final assessment rubric of sustainability competencies were noted.

### 3.5. Common Characteristics of the Pedagogical Approaches Used

As mentioned earlier and as shown in Table 3, the common characteristic of all the *pedagogical approaches* for ESD in initial teacher training are the [30] UNESCO Roadmap recommendations for implementing the Global Action Programme (GAP) on ESD [21]. This programme was developed to help

learners to think critically about their own lifestyles. The GAP calls for a transformation in education and provides opportunities for all learners to acquire the knowledge, skills, values, and attitudes needed to contribute to sustainable development [40]. It is especially aimed at teacher training.

These common characteristic of the *pedagogical approaches* are similar to the key pedagogical approaches in ESD included in the recent report: *Issues and trends in Education for Sustainable Development* [19] (p. 49).

The learning content in the curriculum related to the SDGs (Table 3) includes sustainable consumption (SDG 12) at the three universities and, at UCA, the content also contains good health and well-being (SDG 3), clean water and sanitation (SDG 6), and life on land (SDG 15). All these topics are worked on in a cross-disciplinary manner.

The second characteristic according to the GAP is teaching and learning in an interactive learner-centred way. Project-oriented learning (POL) and problem-based learning (PBL) are two pedagogical strategies recommended for the development of competencies for ESD [30,41]. Those two pedagogical approaches enable the students to work on and do their research projects related to real-world sustainability issues. Through POL and PBL, students work in small groups "as autonomous learners and this emphasizes the active development of knowledge rather than its mere transfer and/or passive learning experiences" [29] (p. 49). The teachers of the subjects in the case studies stimulate and support the reflections of the students and their role was to be facilitators of learning processes [42].

With regard to the third common characteristic of the proposal for ESD in initial teacher training, promote competencies in ESD, the authors and main researchers of this paper selected two competencies of sustainability of the EDINSOST Project, which could be carried out using the specific characteristics of the case studies:

Understands the functioning of natural, social, and economic systems, as well as their interrelations and problems, both at a local and global level (EDINSOST competency no. 1). This competency includes systemic thinking competency [9,10,43]. This competency is meant to promote a holistic view and the relation between the different dimensions of SD.

The second competency in sustainability worked on in this study was: Promotes and participates in community activities that encourage sustainability. (EDINSOST competency no. 3). This competency is similar to "interpersonal competence" [43] and includes the competency for cooperation in heterogeneous groups [10]. It concerns the ability to move from theory to practice, including sustainability values "engaging head, hands, and heart" [44].

## 4. Results

### 4.1. Description of the Sample

In this study, a total of 93 students of the degree in Primary Education participated. They were divided into four groups (in the subject of Didactics of Science: 17 third-year students of the UIC, 13 third-year students of the UCA, 40 s-year students of the US and in the subject of Environmental Education: 23 fourth-year students of the UCA). The results of the assessment of competencies of sustainability and the assessment of the individual EF of the 93 students were gathered. At the beginning of the 2018–2019 academic year, the students calculated their individual EF to have an initial diagnosis of their consumption habits. After a semester, during which different methodological approaches and didactic resources in ESD were implemented in a curricular and extracurricular manner, the students repeated the calculation of their individual EF. The results before and after implementing the proposals for ESD are available for all the participating students. Excel was used to analyze the level of acquisition of competencies and the statistical package SPSS v.24 was used to study the data of the EF and the relation between the EF and the competency level.

### 4.2. Results of the Assessment of Competencies of Sustainability

Through the assessment instrument of two competencies of sustainability (Table 4) and different evaluations (teacher evaluation, evaluation by experts, and self-evaluation), the results of the assessment of competencies shown in Figures 2 and 3 were obtained.

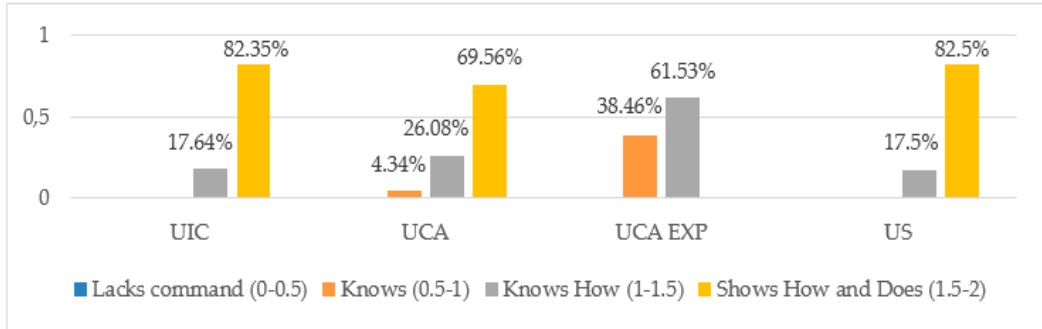

**Figure 2.** SUST 1—Critical contextualization of knowledge establishing interrelationships between local and/or global social, economic, and environmental, problems.

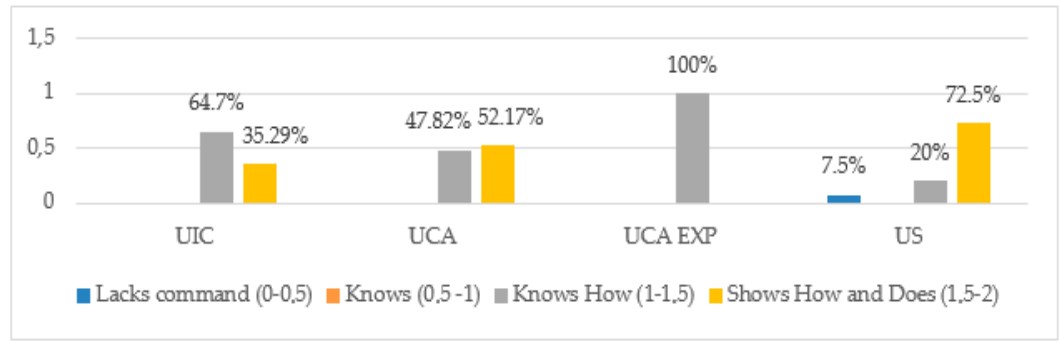

**Figure 3.** SUST 3—Participation in community processes that promote sustainability.

### 4.3. Descriptive Analysis before and after Implementing the Proposals for EDS

Table 5 shows the descriptive statistics of the EF of each group and as a whole before and after implementing the proposals for EDS.

As seen in the results, the global EF is lower after the implementation of the pedagogical approach in the four case studies and therefore in the sum of the four. The students had, to a certain extent, changed their consumption patterns.

Table 6 shows the differences of the variables (before and after) of each group (UIC, UCA_EE, UCA_DofS, US, and total). As can be seen, the differences are positive for all the groups, which indicates that after the implementation, the results improved in all the fractions of the EF except for the UCA_EE and UCA_DofS in which no reduction in the fractions of carbon footprint and goods and services footprint are observed.

When analyzing the case of the UIC, it is observed that there are differences in the results after the didactic implementation in the EF fractions. As the standard deviation shows, the implementation caused a very homogeneous response in the case of housing footprint, that is, the majority of the students gave a similar answer, which did not occur in the rest of the indicators or fractions. The answers are highly heterogeneous in the case of carbon footprint.

**Table 5.** Comparative and joint descriptive statistics (gha) of the Universitat Internacional de Catalunya (UIC), Universidad de Sevilla (US), and Universidad de Cádiz (UCA) groups before and after training.

|  | Variable | Before | After | Standard Deviation before | Standard Deviation after |
|---|---|---|---|---|---|
| UIC = 17 | CF | 1.08 | 0.86 | 0.61 | 0.38 |
|  | FF | 1.50 | 1.38 | 0.27 | 0.28 |
|  | HF | 0.44 | 0.40 | 0.13 | 0.20 |
|  | GSF | 0.87 | 0.78 | 0.24 | 0.23 |
|  | Total | 3.89 | 3.42 | 0.57 | 0.60 |
| UCA_EE = 23 | CF | 0.68 | 0.73 | 0.26 | 0.24 |
|  | FF | 1.65 | 1.64 | 0.48 | 0.27 |
|  | HF | 0.45 | 0.41 | 0.16 | 0.12 |
|  | GSF | 0.77 | 0.67 | 0.28 | 0.22 |
|  | Total | 3.55 | 3.45 | 0.88 | 0.51 |
| UCA_DofS = 13 | CF | 1.05 | 0.91 | 0.56 | 0.48 |
|  | FF | 1.52 | 1.34 | 0.17 | 0.22 |
|  | HF | 0.49 | 0.47 | 0.09 | 0.09 |
|  | GSF | 0.60 | 0.71 | 0.23 | 0.35 |
|  | Total | 3.65 | 3.43 | 0.68 | 0.68 |
| US = 40 | CF | 0.51 | 0.47 | 0.36 | 0.37 |
|  | FF | 1.24 | 1.04 | 0.50 | 0.43 |
|  | HF | 1.06 | 1.01 | 0.60 | 0.58 |
|  | GSF | 0.67 | 0.66 | 0.28 | 0.31 |
|  | Total | 3.47 | 3.17 | 0.66 | 0.71 |
| Total Groups | CF | 0.73 | 0.67 | 0.49 | 0.40 |
|  | FF | 1.43 | 1.29 | 0.46 | 0.42 |
|  | HF | 0.72 | 0.67 | 0.51 | 0.49 |
|  | GSF | 0.72 | 0.69 | 0.28 | 0.29 |
|  | Total | 3.59 | 3.32 | 0.72 | 0.64 |

**Table 6.** Descriptive statistics for the differences before and after the pedagogical implementation in all groups.

| Universities | Variable (Before-After) | N | Minimum | Maximum | Average | Standard Deviation |
|---|---|---|---|---|---|---|
| UIC | CF | 17 | −0.46 | 1.67 | 0.22 | 0.61 |
|  | FF | 17 | −0.36 | 1.31 | 0.12 | 0.40 |
|  | HF | 17 | −0.13 | 0.35 | 0.04 | 0.13 |
|  | GSF | 17 | −0.51 | 0.75 | 0.09 | 0.30 |
|  | Total | 17 | −0.55 | 2.71 | 0.47 | 0.71 |
| UCA_EE | CF | 23 | −0.81 | 0.65 | −0.05 | 0.35 |
|  | FF | 23 | −1.13 | 1.53 | 0.01 | 0.53 |
|  | HF | 23 | −0.26 | 0.46 | 0.04 | 0.17 |
|  | GSF | 23 | −0.93 | 1.06 | 0.10 | 0.36 |
|  | Total | 23 | −1.76 | 3.51 | 0.10 | 0.99 |
| UCA_DofS | CF | 13 | −1.48 | 1.36 | 0.14 | 0.80 |
|  | FF | 13 | −0.79 | 0.30 | 0.18 | 0.30 |
|  | HF | 13 | −0.19 | 0.32 | 0.03 | 0.13 |
|  | GSF | 13 | −0.55 | 0.62 | −0.11 | 0.31 |
|  | Total | 13 | −1.84 | 1.14 | 0.22 | 0.86 |
| US | CF | 40 | −0.48 | 0.62 | 0.03 | 0.19 |
|  | FF | 40 | −0.81 | 1.26 | 0.20 | 0.36 |
|  | HF | 40 | −0.63 | 0.81 | 0.06 | 0.31 |
|  | GSF | 40 | −0.63 | 0.48 | 0.01 | 0.26 |
|  | Total | 40 | −1.37 | 1.68 | 0.30 | 0.69 |
| Total | CF | 93 | −1.36 | 1.67 | 0.06 | 0.45 |
|  | FF | 93 | −1.53 | 1.31 | 0.13 | 0.41 |
|  | HF | 93 | −0.63 | 0.81 | 0.05 | 0.23 |
|  | GSF | 93 | −1.06 | 0.93 | 0.03 | 0.31 |
|  | Total | 93 | −3.51 | 2.71 | 0.27 | 0.80 |

In the case of the UCA group in the Environmental Education subject, the same situation is observed with respect to housing. As mentioned earlier, the negative value for carbon stands out. This indicates that the students' habits related to carbon emissions seem to have worsened after implementing the methodology.

At the UCA in the Didactics of Science subject, the food footprint fraction is the one where the students provided the most homogeneous answers after the methodological implementation and the most heterogeneous response is the carbon footprint, as with the UIC group. The fraction of goods and services is negative, which means the change is the opposite of what was expected after implementing the methodology.

In the case of the US, the most homogeneous answers were given in terms of goods and services and the most heterogeneous ones in terms of carbon footprint, as in the other universities, except in UCA_EE.

When the data of the four cases are analyzed together, it is observed that the housing indicator was answered in the most homogeneous way after implementing the methodology.

The data also reveal that the methodological implementation developed at the UIC had the greatest impact. Although it is not the university that obtained the lowest EF, the difference between the averages is the highest of the three cases, which means the implementation had a greater effect on student behavior.

*4.4. Analysis of the Relation between the Reduction or not of the Ecological Footprint and the Two Competencies of Sustainability*

Table 7 shows how the students evolved according to the level of competency acquired. A relation between high levels of competency in sustainability and EF reduction is observed in Figure 4.

**Table 7.** Cross-tabulation between the different levels of SUST 1 and whether the ecological footprint (EF) decreases or not.

| Count | | Did EF Decrease? | | Total |
|---|---|---|---|---|
| | | Yes | No | |
| SUST 1 levels | Level 1 (0.5–1) | 6 | 2 | 8 |
| | Level 2 (1–1.5) | 21 | 12 | 33 |
| | Level 3 (1.5–2) | 38 | 14 | 52 |
| Total | | 65 | 20 | 93 |

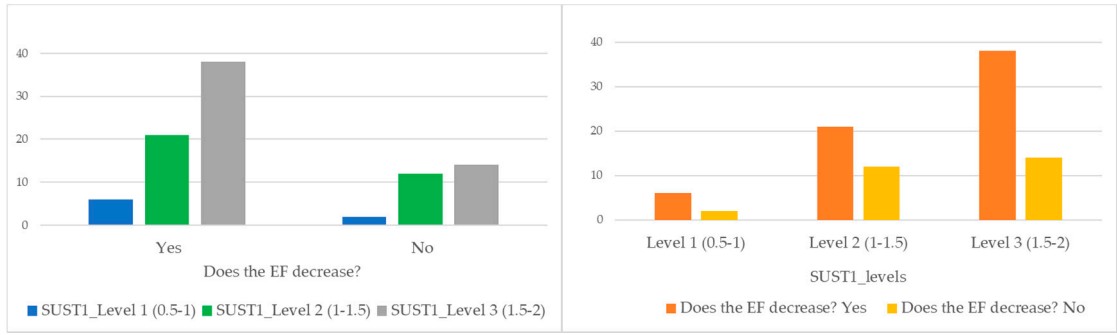

**Figure 4.** EF change in relation to SUST 1.

If the same analysis is performed for SUST 3 in relation to the variable "does the EF decrease", similar results as the ones for SUST 1 are obtained. Table 8 shows there are more students whose EF decreased than those whose EF did not decrease.

**Table 8.** Cross-tabulation between the different levels of SUST 3 and whether the EF decreases or not.

| Count | | Does the EF Decrease? | | Total |
|---|---|---|---|---|
| | | Yes | No | |
| SUST 3 levels | Level 1 (0.5–1) | 3 | 0 | 3 |
| | Level 2 (1–1.5) | 38 | 17 | 55 |
| | Level 3 (1.5–2) | 24 | 11 | 35 |
| Total | | 64 | 28 | 93 |

The greatest EF reduction occurred in the second competency level as in the case of competency 1 and not in the final level. In this case, the changes are not significant (Figure 5).

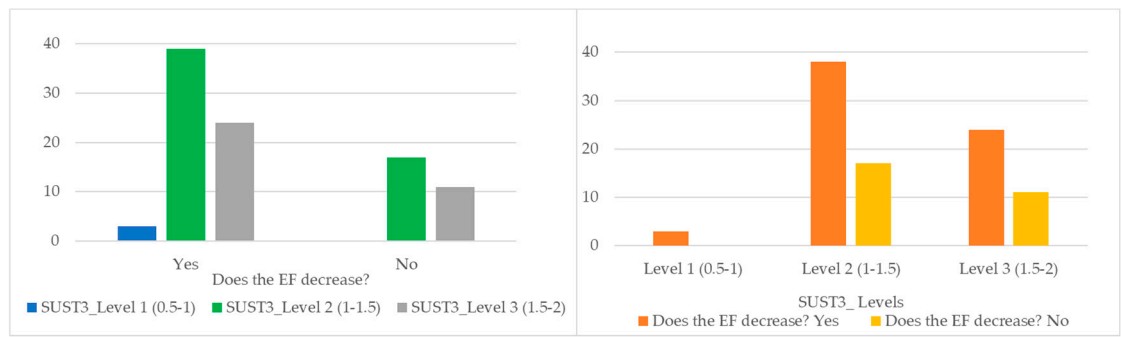

**Figure 5.** EF change and SUST 3.

## 5. Discussion

The main goal of the case studies was to implement pedagogical approaches to ESD in the curriculum in accordance with the Global Action Programme on ESD of future teachers and to measure the students' level of acquisition of competencies of sustainability and their behavior habits. As described in the Methods section, active learning-teaching strategies were used to achieve this aim. In all the case studies, most students reached higher levels of competency acquisition (Figures 2 and 3), which corroborates what ESD experts say: competency development can only be evaluated and achieved by action [42,43,45] and PBL and POL are appropriate strategies for said competency development [46]. Working on real problems or projects that affect students directly helped them associate the different dimensions of sustainability and their integration, as recommended by the 2030 Agenda [5], attaining high levels of SUST 1.

This research focused on the holistic vision of ESD in which, through PBL or POL, interconnections between the different dimensions of sustainability were observed. The study was intended to find out whether the methodological approaches for ESD in teacher training recommended by the GAP on ESD for teachers are effective. The level of acquisition of two competencies of sustainability as well as a possible change in consumption habits was therefore measured. As in almost all behavioral surveys, only environmental variables were measured [26,47,48]. The EF, an instrument tested in HE and more particularly in teacher training [1], and a rubric adapted to teacher training [30] to measure the students' level of acquisition of competencies of sustainability were used in this research.

The other results obtained were analyzed in accordance with the research objectives.

Regarding objective 1, the general results obtained indicate an EF reduction in all the case studies, which means the active teaching-learning strategies implemented contributed to transformative education [29], achieving not only knowledge of sustainability, but also modifying, to a certain extent, the students' consumption measured through their EF [1]. Although this reduction was not significant, it indicates the usefulness of active teaching-learning strategies [21] in real-world problems as suggested by experts [3,10,44]. However, further research is necessary to obtain greater changes. More data are required to analyze the possible causes of these results. In this respect, it would be interesting to

perform a demographic analysis as certain aspects are related to the context of each case and they may have an influence on the results. For instance, as the study concerns students, they do not usually have a steady job and tend to have a limited income. The location of the university and the infrastructure related to public transport may also be determining factors.

With respect to the different EF fractions, the results show a reduction of the EF for all the students in all cases except carbon footprint and goods and services footprint in UCA_EE and UCA_DofS. This could be due to the fact that after implementing the strategies, the students have more knowledge of what the EF implies and answer the final questionnaire with more awareness of the problems in question, since they developed the topics in their subjects. In the case of the Universidad de Cádiz in the subject of Environmental Education, during the study of the socio-environmental problems of the natural park of Toruños, the students did not work on issues related to the carbon footprint fraction much but they carried out an in-depth analysis of the connections between the different socio-environmental aspects of sustainability and the impact of consumption on the environment. It can be considered a success that the students demonstrated a reduction in the goods and services fraction. The relationship between theory and practice [46] contributed to the development of more sustainable habits. In the case of the subject of Didactics of Science at the same university, the carbon and food footprint fractions decreased, but the goods and services fraction increased. This may be due to the fact that in this subject, besides working on SDG 12, which may have influenced the decrease in the carbon footprint fraction, they also addressed SDG 3 and SDG 6, emphasizing the relationship between sustainable food, water, and health. However, they did not deal with aspects such as spending and saving habits or recycling, the main dimensions of the goods and services fraction.

Finally, it is worth noting that, for the carbon footprint fraction, except in the case of the Universidad de Cádiz in the Didactics of Science subject, the strategies used caused greater dispersion in the answers of the students and, therefore, greater diversity. This may be due to the fact that the specific content worked on in each subject did not focus on the same sustainability issues. The opposite occurred for the housing fraction, where the results became more homogeneous, possibly because the students do not usually change their homes in a four-month period and it is therefore logical that there are hardly any changes with respect to the initial situation. The analysis of the results shows the difficulties involved in changing habits of human behavior [49] and the time it takes for those changes is significant.

The second research objective was to analyze whether there is a relation between high levels of competency acquisition in sustainability and the reduction of the individual EF. The results show that there is indeed a relationship between the high levels of competency acquisition in sustainability (Knows How and Shows How and Does) and EF reduction, although the data, as in the previous case, are not significant. Differences are also observed between competencies 1 and 3. As shown in Figure 2, for SUST 1, which refers to the interrelation between the different dimensions of sustainability, the greatest EF reduction is concentrated in level 3 (Shows how & Does): is able to imagine and predict the impacts the changes produced in natural systems may cause in social and economic systems and among each other. It seems logical that the students who acquired this level of commitment modify their consumption habits by reducing their EF.

Regarding SUST 3: Participation in community processes that promote sustainability, Figure 3 shows that there is a greater correlation between EF reduction and level 2: Is able to interact satisfactorily in educational community projects, encouraging participation. The reason for this could be that in the active teaching-learning strategies used in all the case studies, issues and projects were worked on that made the students get out of the university to interact with other people and communities in a participative manner. It is not so much a transformation of reality, but a transformation of the students themselves. Usually, in educational processes, even in active ones, more time is devoted to analyzing problem situations and establishing connections between socio-environmental issues than to direct action. Direct action occurs at the end of the process since at the beginning it is essential to establish the bases for action to make sense and to be useful for the community it is meant for. Transformative

education took place, but for carrying out socio-educational activities in participatory communities, the length of participation time needs to be longer than just a few months.

## 6. Conclusions

This comparative study shows that there are different ways to implement ESD in the initial teacher training curriculum. When using active teaching-learning strategies related to the SDGs in cross-curricular contexts, competencies of sustainability are developed that enable future teachers to face current and future sustainability challenges.

EF is a highly useful tool for working with students as it enables converting personal behavior into quantitative data, which gives students a clear image. It is a space in which one can critically reflect on the consequences of daily actions as consumers and analyze the connections between the environment and the other dimensions of sustainable development. The EF study allowed measuring in a quantitative manner to what extent there have been advances towards more conscious and sustainable consumption and its relationship with the level of sustainability competency acquisition, as well as what aspects should be stressed in future studies.

Active teaching-learning strategies contribute to transformative education since there is a reduction in the EF and higher levels of sustainability competency acquisition are achieved. However, it is necessary to implement pedagogical approaches on ESD more generally and at different levels if the aim is to cause important changes in behavior. The results obtained in this study are not significant, probably because the pedagogical approaches were only used for short periods of time in the different subjects.

## 7. Limitations of the Study

The main difficulty of this study is the short period (a four-month period in a single academic year) in which the teaching-learning strategies were implemented. As it was not compulsory for the students to answer the EF questionnaire, the sample does not include the total number of students of the subjects, but only those who provided their EF data at the beginning and at the end of the implemented pedagogical approaches. The training acquired in the subjects contributed to modifying the students' level of awareness when filling out the EF questionnaire at the end of the subject.

**Author Contributions:** Conceptualization, Methodology, Validation, Investigation, Resources, Writing–Review & Editing, Visualization, S.A.-T., E.G.-G., R.J.-F. and C.S.-E.; Data Curation, Writing Original Draft Preparation, Project Administration, S.A.-T.

**Funding:** This research was funded by the EDINSOST Project "Education and Social Innovation for Sustainability. Training at Spanish universities of professionals as agents of change to face the challenges of society", funded by the "R&D State programme oriented towards the challenges facing society" of the Spanish Ministry of Economy and Finance [Ref. EDU 2015-65574-R] and also by the Tatiana Pérez de Guzmán el Bueno Foundation.

**Acknowledgments:** The authors gratefully acknowledge the support of the Integral Sustainability and Education (SEI) Research Group of the Universitat Internacional de Catalunya (2017 SGR 119). The authors also kindly thank M. Yolanda Fernández from the Universidad Europea Miguel de Cervantes in Valladolid, Spain, for the statistical advice she provided.

**Conflicts of Interest:** The authors declare no conflict of interest.

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
