# Peer review of "Implementing Pedagogical Approaches for ESD in Initial Teacher Training at Spanish Universities"

_sustainability, doi:10.3390/su11184927_

Round 1
Reviewer 1 Report
This paper addresses the important topic of Higher Education pedagogies and their potential to promote competencies in sustainability in general and in sustainable consumption in particular. Three different methodologies are analyzed in the context of three Spanish universities. Effectiveness of the pedagogies is assessed based on competence acquisition measures. With this outset, the paper can be considered relevant for the field of Higher Education for Sustainable Development. In its current form, however, I see some minor and major shortcomings in the manuscript that the authors may consider in the revision of the paper.
Major issues:
- While I appreciate the broader stage setting with the timely references to FFF, the SDGs and the broader need for sustainability transformations, I would like to see a deeper background section contextualizing the paper in the different academic discourses this paper is connected to (teacher education for sustainable development, competence assessment, teaching and learning strategies, sustainable consumption). In its current version, the paper jumps straight into the empirical study after a short general introduction. I suggest to include a background section that talks about competence development in teacher education for sustainable development, education for sustainable consumption as a specific strand in (T)ESD, and the state of research on the effectiveness of different pedagogical/curricular approaches to promote competence development and sustainable consumption behaviors.
- There is a rich debate on active learning methodologies and sustainable consumption teaching that is not recognized in this paper. Here are some references that the authors may want to review to broaden and enrich the background and discussion section of the paper:
Merritt, E., Hale, A., & Archambault, L. (2019). Changes in Pre-Service Teachers’ Values, Sense of Agency, Motivation and Consumption Practices: A Case Study of an Education for Sustainability Course. Sustainability, 11(1), 155. https://doi.org/10.3390/su11010155
Fischer, D., & Barth, M. (2014). Key Competencies: Reconciling Means and Ends in Education for Sustainable Consumption. In V. Simovska & P. M. McNamara (Eds.), Schools for Health and Sustainability: Theory, Research and Practice (pp. 41–60). Dordrecht: Springer.
Sahakian, M., & Seyfang, G. (2018). A sustainable consumption teaching review: From building competencies to transformative learning. Journal of Cleaner Production, 198, 231–241. https://doi.org/10.1016/j.jclepro.2018.06.238
Frank, P., & Stanszus, L. S. (2019). Transforming Consumer Behavior: Introducing Self-Inquiry-Based and Self-Experience-Based Learning for Building Personal Competencies for Sustainable Consumption. Sustainability, 11(9), 2550. https://doi.org/10.3390/su11092550
Barth, M., Adomßent, M., Fischer, D., Richter, S., & Rieckmann, M. (2014). Learning to change universities from within: A service-learning perspective on higher education for sustainable consumption. Journal of Cleaner Production, 62(1), 72–81. https://doi.org/10.1016/j.jclepro.2013.04.006
The Active Learning Methodology series of the Partnership of Education and Research on Responsible Living (PERL), run by UNESCO Chair Victoria Thoresen.
- The paper lacks consistency with regard to its use of terminology. Environmental education, education for sustainable development and education for sustainability are seemingly used interchangeably (while for many scholars in the field represented significantly different approaches). What is maybe even more crucial is that the authors do not seem to clearly distinguish between "environmental" and "sustainable". What term reflects the scope of the paper? What is the authors' understanding of this term? The title for example speaks of sustainable consumption, but what is measured in the end is the ecological footprint, which accounts only for the ecological impacts of consumption, but not socio-economical impacts. How sensitive is this tool to short term changes (caused, e.g., by an educational intervention)? At least a more critical discussion of this indicator would be helpful (the authors say that "it has limitations", p. 4, but do not expand on this any further).
- I have some major concerns about the methodology. In technical regards, I miss key statistical information that helps me to understand and comprehend the authors' argument and interpretation (minimum / maximum of scales, chi-square test parameters etc.). Please refer to the APA standards for reporting these kinds of information. Also, there is little or no information on the instruments used: how were the scales constructed? Some more information on what the footprint calculator measures would be helpful. More information on the competence assessment tool is essential. Given the plethora of works struggling to assess competencies in the sustainability space, this is a crucial piece of information that is still missing in this paper. Please provide more information on how the assessment tool was constructed, its key statistical indicators, how students were categorized into different levels, and some exemplary items, so that other researchers are able to critically appraise your approach.
- Overall, the research design does not lend itself to make causal claims about differences in pre and post, given the small numbers and lack of adequate control groups. The language in the results, discussion, and conclusion part of the paper however suggests that the research allows to draw these kinds of conclusions. Please revisit the manuscript and these sections in particular for these occurences.
Minor issues:
- Overall, the authors use acronyms heavily throughout the paper. While I see that this is necessary when several variables are used, as a reader I felt myself browsing back and forth trying to identify what which acronym meant. I'd like to encourage the authors to think of alternative ways to keep the information concise and be more accomodating to readers.
- I am not sure about the use of the term "methodology", as for many researchers this refers to more fundamental questions of research paradigms, assumptions, and approaches. What you have implemented (if I understand correctly) are different teaching and learning settings that have been designed on the basis of different pedagogies. There is no right or wrong here, but I feel that it may be easier for many readers to connect to a more "educational" language here.
- The first research objective reads: "Implement sustainability in the university curriculum of future teachers and measure the students’ level of acquisition of competencies in sustainability." Is the curricular implementation of sustainability really a *research* objective and object of investigation?
- No need to mention the statistical package that was used to analyze data in the abstract.
I wish the authors all best for the revision!
Reviewer 2 Report
The introduction is best started around the discussion point of the SDGs. The overall discussion of the issues associated with sustainability literature in the context of university education and degree programmes is currently very weak and needs to be substantially improved. There is a substantial history on transformative learning and sustainability and environmental education this needs to be discussed. This needs to be clearly dealt with in both the introduction and in a substantial literature review.
The problem in the context of the paper should be in terms of sustainability education and its effectiveness not the "crisis" as that has long been recognised.
In terms of the case studies there needs to be a clearer account of exactly what they were focused on - please provide more detail.Make sure all SDGs are clear, provide details on UNESCO roadmap and other reports that way the content of the case studies will make more sense to readers.
The research only has a small sample size, please discuss other factors that may influence results - such as demographics. Given the number of participants please also provide a stronger justification of statistical methods and the conclusions that can be drawn.
Re ethics - the study should briefly note the ethics procedures/permissions received
Finally, the paper should be retitled as the paper does not actually have a strong grounding in methodology
Round 2
Reviewer 2 Report
thank you for your efforts with the paper, while I still tjhink that the contextualisation needs work, as does the research design, the paper has advanced to a stage wher it will be of interest to some readers in the education field.